# CO$_2$ Decomposition in Microwave Discharge Created in Liquid Hydrocarbon

Timur S. Batukaev [1], Igor V. Bilera [1], Galina V. Krashevskaya [1,2], Yuri A. Lebedev [1,*] and Nurlan A. Nazarov [1]

[1] A.V. Topchiev Institute of Petrochemical Synthesis of the Russian Academy of Sciences (TIPS RAS), Leninsky Ave. 29, Moscow 119991, Russia
[2] LAPLAZ, National Research Nuclear University MEPhI, Kashirskoe Shosse, 31, Moscow 115409, Russia
[*] Correspondence: lebedev@ips.ac.ru

**Abstract:** The task of CO$_2$ decomposition is one of the components of the problem associated with global warming. One of the promising directions of its solution is the use of low-temperature plasma. For these purposes, different types of discharges are used. Microwave discharge in liquid hydrocarbons has not been studied before for this problem. This paper presents the results of a study of microwave discharge products in liquid Nefras C2 80/120 (petroleum solvent, a mixture of light hydrocarbons with a boiling point from 33 to 205 °C) when CO$_2$ is introduced into the discharge zone, as well as the results of a study of the discharge by optical emission spectroscopy and shadow photography methods. The main gas products are H$_2$, C$_2$H$_2$, C$_2$H$_4$, CH$_4$, CO$_2$, and CO. No oxygen was found in the products. The mechanisms of CO$_2$ decomposition in the discharge are considered. The formation of H$_2$ occurs simultaneously with the decomposition of CO$_2$ in the discharge, with a volumetric rate of up to 475 mL/min and energy consumption of up to 81.4 NL/kWh.

**Keywords:** microwave discharge in liquid hydrocarbons; CO$_2$ decomposition; chromatography of discharge products; optical emission spectroscopy; shadow photograph; hydrogen production





## 1. Introduction

In recent years, much attention has been paid to the problem of global warming. One part of this problem is related to CO$_2$ emissions into Earth's atmosphere. CO$_2$ emissions result from the combustion of fossil fuels (coal, natural gas, and oil) in energy and transport. An urgent problem, addressed by many researchers, is the problem of reducing the emissions and use of CO$_2$. One way to solve the problem is to use CO$_2$ as a feedstock for obtaining useful products, in particular, for the industrial production of synthetic fuels and chemical products [1–6].

Another way is the decomposition of CO$_2$. Currently, a lot of research is being carried out on energy efficient CO$_2$ conversion technologies, such as thermal catalysis, electrocatalysis, photocatalysis, bioelectrocatalysis, etc. (see, for example, [7,8]). Among them, a special place is occupied by the method of plasma-chemical decomposition of CO$_2$, using a weakly ionized low-temperature plasma of gas discharges, and hybrid methods with the simultaneous use of plasma and catalytic/photocatalytic materials [9–14].

Various types of gaseous electrical discharges are used to decompose CO$_2$, such as DC discharges, RF and microwave discharges, and barrier discharges. There is practically no information about such processes being implemented in a microwave discharge in liquid hydrocarbons [15]. At the same time, this discharge has a number of advantages in relation to the CO$_2$ decomposition process. First, the discharge is surrounded by a liquid, whose vapor enters the discharge. The temperature of the liquid does not exceed the boiling point of the liquid. This ensures a high rate of quenching of the products (in the case of CO$_2$ decomposition, this is important to suppress the reverse reactions leading to the formation of CO$_2$). Second, it is known that the addition of methane and hydrogen to the discharge

with $CO_2$ improves the decomposition of $CO_2$. The positive effect of $CH_4$ additions on the degree of $CO_2$ conversion was observed experimentally in a barrier discharge [16–18], in a gliding arc reactor [19–24], in a spark discharge [25], in a glow discharge [26], and also in a rotating arc [27] and in thermal plasma [28]. The addition of $H_2$ also increases the degree of $CO_2$ decomposition [29,30]. It has also been shown that such additives contribute to the removal of oxygen from the mixture of products, with the formation of water.

Hydrogen and methane are the molecules that are produced in microwave discharges in liquid hydrocarbons [15]. All this arouses interest in experiments on the decomposition of $CO_2$ in a microwave discharge in liquid hydrocarbons.

This paper presents the results of studying the degree of $CO_2$ decomposition, and the characteristics of a microwave discharge in a liquid hydrocarbon during $CO_2$ bubbling.

## 2. Experimental Setup

The experimental setup used for generation of a microwave discharge in liquid hydrocarbons was described in detail in [31,32]. It includes a microwave (2.45 GHz) magnetron generator, circulator, water attenuator, directional coupler, spectrum analyzer, and an oscilloscope. The attenuator allows one to have a smoothly varying incident power, in the range from 100 W to 2.5 kW. The discharge section is a waveguide-to-coaxial junction, the center conductor of which serves as an antenna for introducing microwave energy into the reactor. The discharge was created in liquids near the tip of the antenna in a quartz reactor (diameter 55 mm) filled with liquid hydrocarbon, and placed within a protective metal screen. The antenna part coming out into the reactor is made of stainless steel tubing, with an external diameter of 2.0 mm.

Schematically, the discharge chamber and the measurement tools are shown in Figure 1. The volume of liquid hydrocarbon in the reactor is about 40 mL, which ensures that the end of the internal electrode of the coaxial line is located below the surface of the liquid. Experiments were carried out with an incident microwave power of about 200–350 W. The pressure above the surface of the liquid was equal to atmospheric pressure. The duration of the experiment was about one minute. During this time, the gas flow rate at the reactor outlet was measured, and a gas sample was taken to analyze the composition of the products. During one experiment, the composition of the liquid in the reactor was not updated. Figure 1 includes a photograph of the discharge.

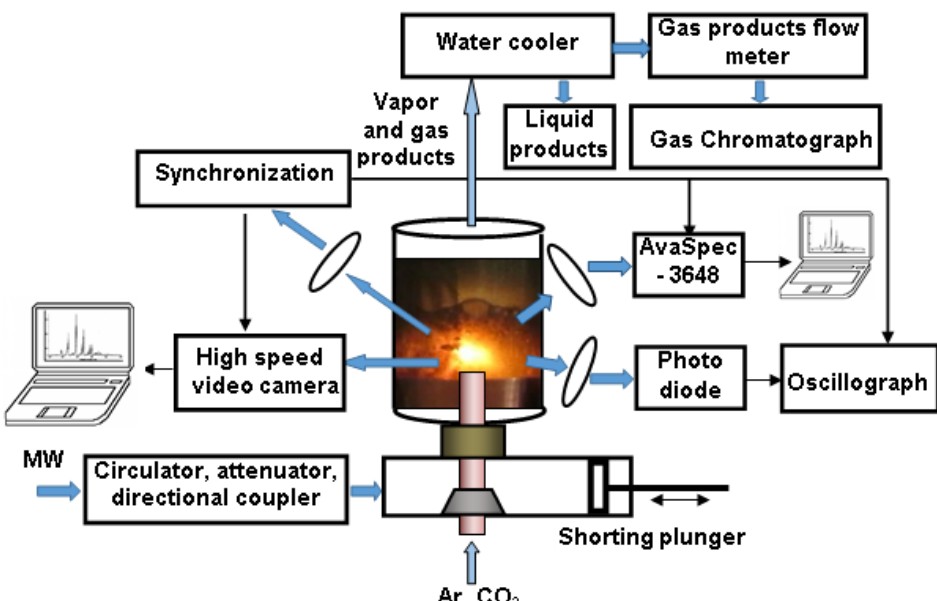

**Figure 1.** Schematic of the experimental setup.

The discharge was investigated by optical methods in the visible wavelength range (Figure 1). The change in the total intensity of radiation, integrated over the spectrum and volume of the discharge, was investigated using a photodiode for visible light, BS520, connected to an AKIP-4126/3A-X oscilloscope. The time resolved emission spectrum of the discharge, in the wavelength range 172–744 nm, was investigated using an AvaSpec-3628 gated spectrograph, with a spectral resolution of 1 nm. The discharge, through optics, was focused on the input aperture of the optical fiber, which directed radiation to the entrance slit of the spectrometer. The relative calibration of the spectrograph was carried out using a band tungsten lamp, SI-8-200. The spectra were recorded with a shutter speed of $\geq 1$ ms. Measurements with a shorter exposure were hampered by the low intensity of the discharge radiation. The measured spectra were processed using the Specair-3.0 program [33].

The discharge was visualized with a time resolution using a 9-frame nanosecond video camera, K-011. When obtaining shadow photographs of the discharge, a white light-emitting diode (LED) was used as a source of illumination.

All devices were synchronized with the moment of discharge ignition using an optical synchronization sensor (when a discharge occurs, the sensor issues a rectangular pulse with a duration of 5 µs and a rising edge of 40 ns). All devices were triggered by the rising edge of the sync pulse.

Measurements were performed using petroleum solvent Nefras S2 80/120 as a liquid medium. Petroleum solvent Nefras S2 80/120 is a mixture of light hydrocarbons, with boiling temperatures ranging from 80 to 120 °C. This solvent can be considered as representative of nonpolar liquid alkanes ($C_nH_{2n+2}$, $n < 8$). Our previous study showed that the emission spectra and composition of the main gas products in Nefras S2 80/120 were the same as for other nonpolar hydrocarbons [34,35]. Note that nonpolar hydrocarbons are transparent in the visible wavelength range, which is important for carrying out optical measurements, which are practically the only method for diagnosing such discharges. In the process of burning a discharge in hydrocarbons, solid particles appear in a liquid, which reduces its transparency.

It should be also noted that the dielectric constant is approximately the same for nonpolar liquid hydrocarbons ($\varepsilon \sim 2.0$), and the tangent of losses for nonpolar hydrocarbons is of the order of $10^{-4}$, and the loss of microwave energy to heating the liquid is small. This means that the electrodynamic properties of a microwave plasma generator are practically independent of the type of nonpolar hydrocarbon used. In the process of burning a discharge in liquid hydrocarbons, one of the products is a solid carbon-containing phase, which naturally increases the loss tangent.

To suppress the process of soot formation, the experiments were carried out with argon bubbling through a channel in the antenna. In the experiments, the argon flow was mixed with $CO_2$. The flow rate of the gas mixture was kept constant, at 0.6 L/min, the $CO_2$ flow rate was 10% and 4.7% of the total flow rate (60 and 28.2 mL/min).

A water cooler was used to separate the products of the plasma-chemical reactions from the evaporated hydrocarbon and liquid products. Using a flow meter, the flow rate of the initial gas mixture, and the flow rate of the gas mixture during the burning of the discharge, were controlled.

The content of $CO_2$ in the initial mixture, as well as the composition of the main gas products at the outlet of the reactor with a discharge, were determined by analysis on a portable gas chromatograph with back flush PIA (NPF MEMS, Samara, Russia). Preliminary studies have shown that the main gaseous products in the microwave discharge, in all investigated liquid hydrocarbons, are $H_2$, $C_2H_2$, $C_2H_4$, $C_2H_6$, and $CH_4$ [35]. Since it is supposed to study the decomposition of $CO_2$, the gas mixture may contain $CO_2$, CO, and $O_2$. All these components were determined by a gas chromatograph with two chromatographic columns with Hayesep N adsorbents and 13F molecular sieves. The carrier gas was argon.

### 3. Results and Discussion

The obtained results are illustrated in the figures presented in this part of the paper. It should be noted here that all the energy dependences in the article will be given as a function of the incident microwave power. This approach requires some explanation. For a correct description of plasma processes, the energy characteristic should be the power absorbed by the plasma (or the specific absorbed power). Determination of this quantity in microwave discharges is associated with great difficulties. Even if the discharge section is well matched to the generator, or a reliable measurement of the reflected power is ensured, the power value corrected for this cannot be attributed to the power absorbed in the plasma. This is due to uncontrolled energy losses in the device providing energy supply to the reactor, and losses in the reactor itself. Such energy losses can reach 50% [36]. Therefore, for the energy characterization of the results in the present work, a reliably measured incident microwave power is adopted. It should be understood that, in this case, the real energy efficiency of the processes could be underestimated by 2–3 times.

#### 3.1. Results of Optical Diagnostics

Discharge was studied by the method of optical emission spectroscopy, using the AvaSpec-3628 spectrometer, and the method of shadow photographs using a high-speed video camera.

The emission spectra of microwave discharges in liquid Nefras S2 80/120, with bubbling of an Ar + $CO_2$ gas mixture, contain strong Swan bands (transition $C_2(d^3\Pi_g - a^3\Pi_u)$). Sequences with $\Delta v = 0$ (maximum of emission at 516.5 nm), $\Delta v = 1$ (maximum at 563.5 nm), $\Delta v = -1$ (maximum at 473.75 nm) were used for the determination of rotational and vibrational temperatures, using the Specair code [34]. Figure 2 illustrates good agreement between the measured and calculated spectra for these sequences. Following [35], the rotational temperature of the $C_2$ molecule can be identified with the gas temperature under the experimental conditions.

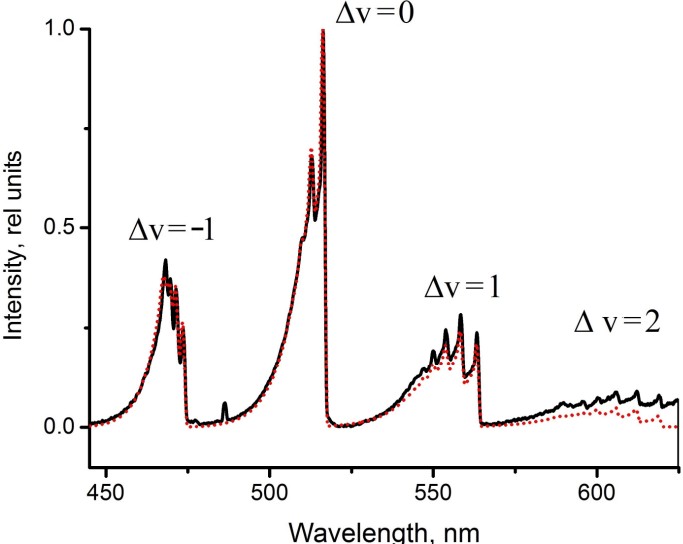

**Figure 2.** Comparison of measured (black solid line) and calculated (red dot line) spectra of Swan bands.

Since the liquid was not renewed during the experiment, it can be expected that the properties of the plasma may change with time. This fact was noted in [37]. Figure 3 shows the emission spectra of the discharge in the wavelength range of 400–600 nm, in which the Swan bands obtained at different times from the ignition of the discharge are emitted. The spectra were recorded with an interval of 2–3 s. The figure does not show all the measured spectra, but only typical ones, reflecting the dynamics of their change. When the discharge is ignited (Figure 3, curve 1), there is no broadband continuum in the

spectrum, associated with solid particles formed in the plasma [37]. As particles form, a continuum appears (Figure 3, curve 2), and with further discharge burning, it is they that determine the discharge emission spectrum (Figure 3, curve 3). The latter occurs 10–15 s after the ignition of the discharge.

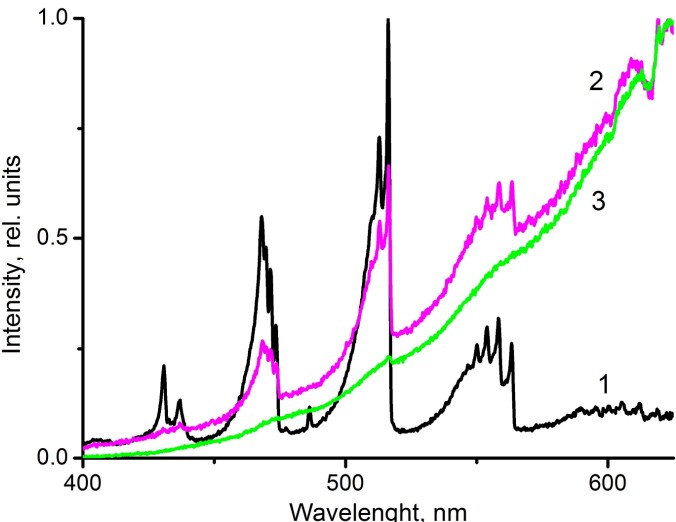

**Figure 3.** Dynamics of a spectrum during an experiment. Numbers near the curves correspond to the spectra measured at different times from the discharge ignition (1-2-3 indicate evolution of spectrum).

When the discharge burns, the ratio of the rotational and vibrational temperatures of the $C_2$ molecule changes. This is illustrated in Figure 4. The time interval between the displayed measurement points is about 2 s. It can be seen that, if at the beginning there is a significant difference between these temperatures, then after 10 s, vibrational–rotational equilibrium sets in.

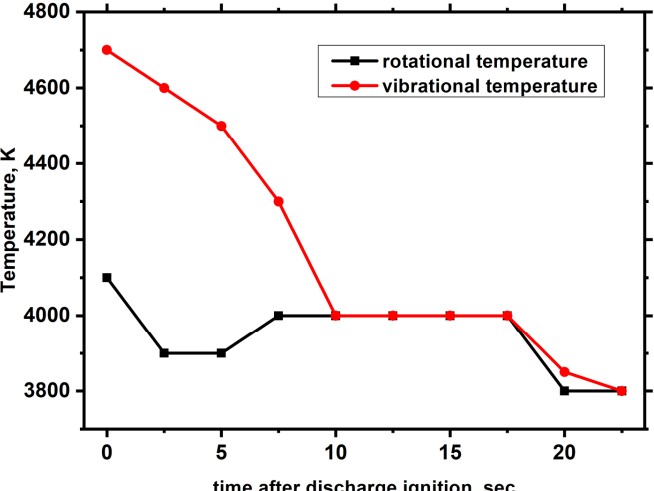

**Figure 4.** Changes in rotational and vibrational temperatures over time during discharge burning in an argon and $CO_2$ gas mixture, with flow rate of 675 mL/min, a $CO_2$ concentration of 5.27%, and an incident power of 200 W.

It is known that microwave discharges in liquids burn in gas bubbles created during liquid evaporation in the region of maximum microwave field strength [15]. In our case, this is the region at the end of the antenna. Figure 5 shows the dynamics of gas bubble development after discharge ignition. The black rectangle at the bottom of each photo corresponds to the antenna. The bright area at the end of the antenna is the glowing area

of the discharge. It is surrounded by a black area, which is an image of a gas bubble, the outer size of which corresponds to its diameter. The luminous region outside the bubble is formed by radiation from an external light source.

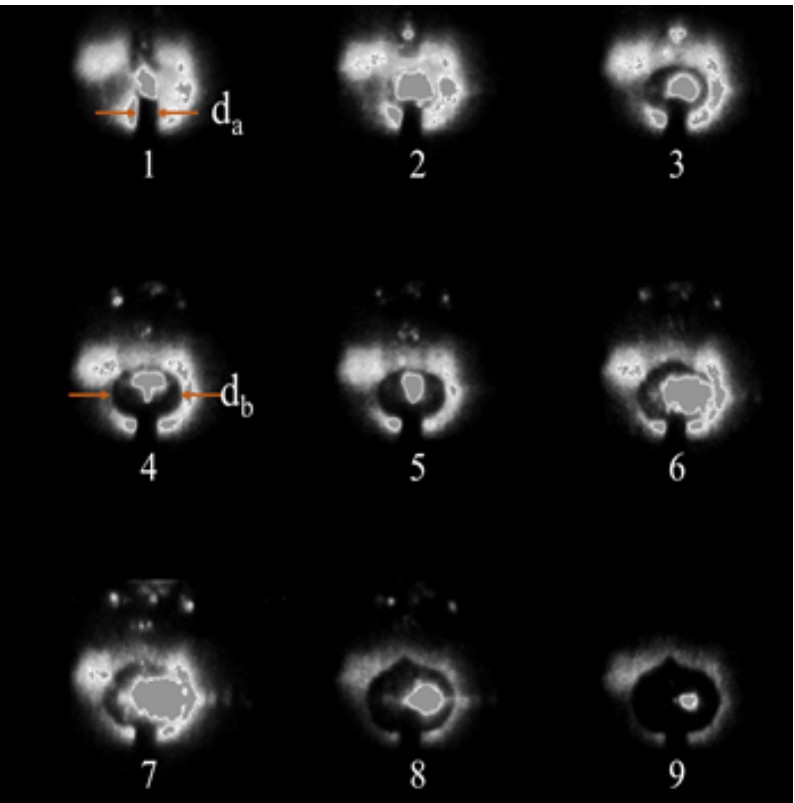

**Figure 5.** Successive set of nine shadow photographs of the discharge with the frame duration of 0.1 µs and spacing of 500 µs. The diameter of the plasma bubble ($d_b$) can be obtained by comparing it with the diameter of the cylindrical antenna ($d_a$) shown on the bottom of each photograph, which is 2 mm.

Shadow photographs make it possible to determine the diameter of the bubble and its growth rate. The final bubble diameter reaches a size of about 1 cm, and the diameter growth rate is about 2.5 m/s. In Figure 5, the image of the bubble has the shape of an ellipse. This is because the discharge is illuminated through a cylindrical lens, which is a discharge tube with liquid, the axis of which coincides with the axis of the cylindrical antenna.

The obtained dimensions allow us to estimate from above, the specific energy input into the plasma from the incident power, which is in the range of 150–350 W/cm$^3$. As already noted, the real energy input into the plasma could be 2–3 times less. These figures are useful for estimating the energy efficiency of processes in the discharge.

### 3.2. Gas Products on the Outlet of Reactor

Chromatographic analysis of the microwave discharge products at the reactor outlet in liquid Nefras S2 80/120, when $CO_2$ diluted with argon is introduced into the discharge zone, showed that the main components of the mixture are $H_2$, $C_2H_2$, $C_2H_4$, $CH_4$, $CO_2$, and CO. No oxygen was found in the products. Figure 6a–d shows the volumetric production rates of the main products at the reactor outlet. These values were obtained from the measured rate of the total gas flow at the outlet of the reactor, taking into account its composition, measured by the chromatographic method.

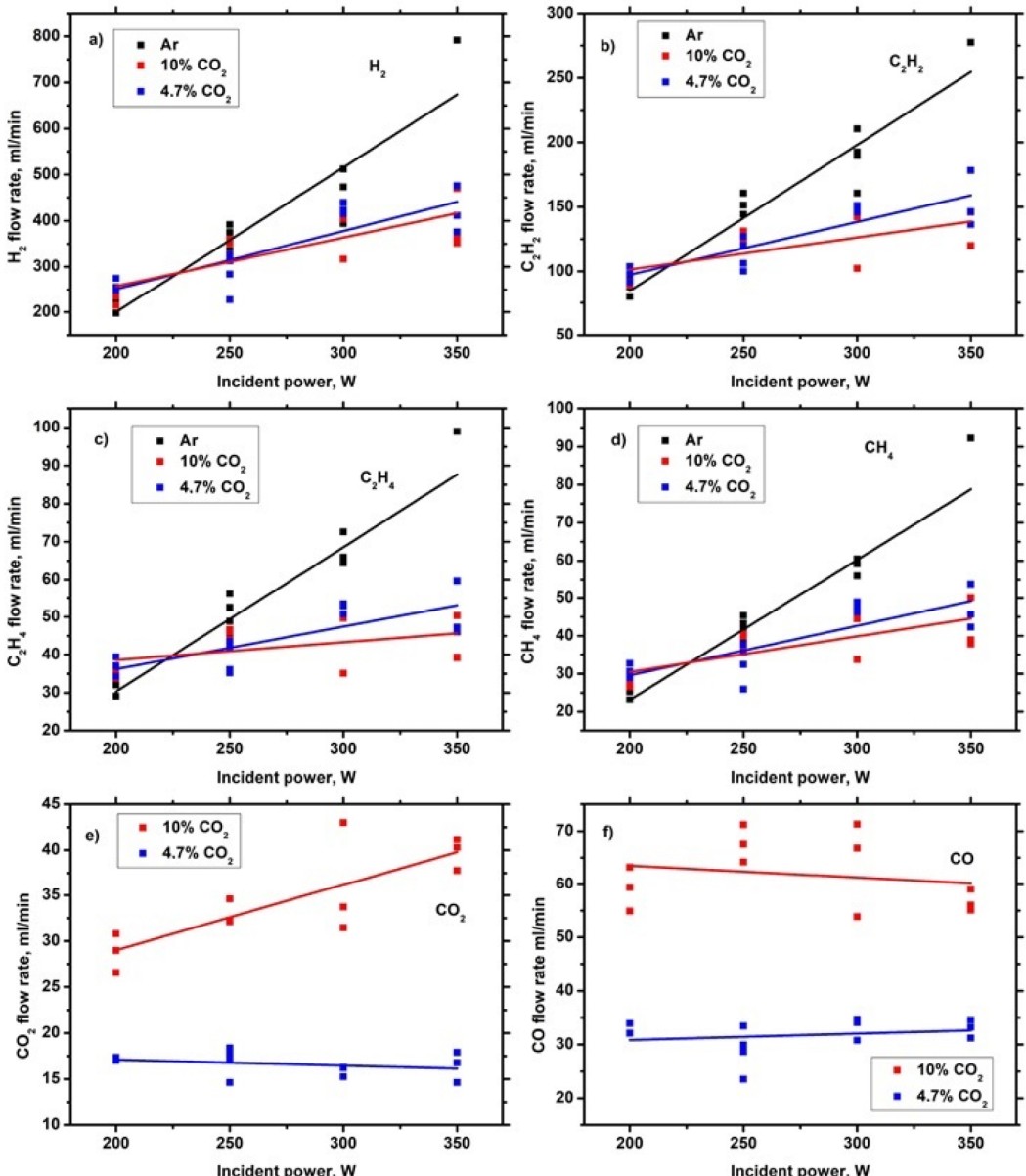

**Figure 6.** Flow rates of gas products in the outlet of the reactor: $H_2$ (**a**), $C_2H_2$ (**b**), $C_2H_2$ (**c**), $CH_4$ (**d**), $CO_2$ (**e**), CO (**f**).

Figure 6a–d shows that the volumetric rates of formation of hydrocarbon products and hydrogen, with the addition of $CO_2$, as a rule, are lower than the rates obtained without the addition of $CO_2$.

The reason for the decrease in the concentration of gaseous products may be as follows. The products of microwave discharges in liquid hydrocarbons, in addition to gas products, are solid carbon-containing particles, with a content of C/H~1–2 [15]. In this case, the bubbling of molecular gases practically does not affect the efficiency of the solid phase formation process. To suppress this process, it is necessary to introduce inert gases, in particular argon, into the plasma [38]. Under the conditions of the experiments, the total flow rate of the mixture of bubbling gases remained unchanged. With an increase in the flow rate of $CO_2$, and a constant total flow rate, the concentration of argon in the plasma decreases and this contributes to an increase in the fraction of solid particles and a decrease in the fraction of gaseous products in the formed products. This follows from the results of [37].

Attention is drawn to the growth of the $CO_2$ velocity at the reactor outlet when the $CO_2$ consumption at the inlet is 10% of the total mixture flow, and the drop in the CO yield (Figure 6e,f). This means that the degree of $CO_2$ decomposition in this case decreases with increasing incident power. At the same time, at a lower $CO_2$ supply rate, its output velocity decreases. This behavior of product yields can be associated with a change in the residence time of $CO_2$ in the discharge region. It is known that with an increase in the incident power in a microwave discharge in liquid hydrocarbons, the lifetime of a bubble with plasma on the antenna decreases [31]. This means that the residence time of $CO_2$ in the discharge zone decreases and, accordingly, the degree of decomposition decreases. According to [31], the plasma bubble lifetime near the antenna is about 1 ms. Estimates show that at a $CO_2$ consumption of 10% of the total mixture flow, the time of flight of $CO_2$ molecules through the discharge is about 1 ms (the characteristic size of the discharge region was obtained from Figure 5). Thus, these times are comparable, and the effect can be observed. The greater the incident power, the shorter the lifetime of the discharge region and the shorter the residence time. At a lower $CO_2$ flow rate, there is no decrease in the discharge lifetime (Figure 6e,f).

Let us consider possible mechanisms of decomposition of $CO_2$ in plasma, which will make it possible to explain the results shown in Figure 6. If we assume that $CO_2$ decomposition proceeds in accordance with the following reaction [38]:

$$CO_2 \rightarrow CO + 0.5O_2, \tag{1}$$

then, as follows from Figure 6e,f, at low incident powers an excess of carbon oxides is observed in the products, in comparison with reaction (1). This may mean, in particular, that the oxygen formed in reaction (1) interacts with the hydrocarbon decomposition products of Nefras S2 80/120 to form CO. As follows from Figure 6f, the number of CO molecules at the outlet of the reactor, at least at low incident powers and a $CO_2$ content of 10%, is equal to twice the amount of decomposed $CO_2$.

It is known that at temperatures exceeding 1000 °C (under experimental conditions, the temperature exceeds this value, see Section 3.2), the reaction between $CO_2$ and $CH_4$ proceeds to form $H_2$ and CO [39]:

$$CO_2 + CH_4 \rightarrow 2CO + 2H_2. \tag{2}$$

If we accept that the decomposition of $CO_2$ under the experimental conditions proceeds according to reaction (2), then it explains the number of CO molecules at the outlet of the reactor. It also explains the absence of oxygen in the reaction products. In addition, the hydrogen formed in reaction (2) increases the yield of hydrogen compared to the case without the addition of $CO_2$.

Thus, it can be assumed that it is reaction (2) that determines the process of $CO_2$ decomposition when $CO_2$ is bubbling through a microwave plasma in liquid hydrocarbons at low incident powers. At low $CO_2$ flow rates, this is observed in the entire range of the studied incident powers. At high flow rates of $CO_2$, and an increase in the incident power, the considered mechanism may be violated due to a decrease in the residence time of $CO_2$ in the plasma region.

### 3.3. CO2 Decomposition and Production of Hydrogen

Figure 7 shows the dependence of $CO_2$ decomposition degree on the microwave incident power. An explanation of the dependencies in Figure 7 is given in Section 3.2.

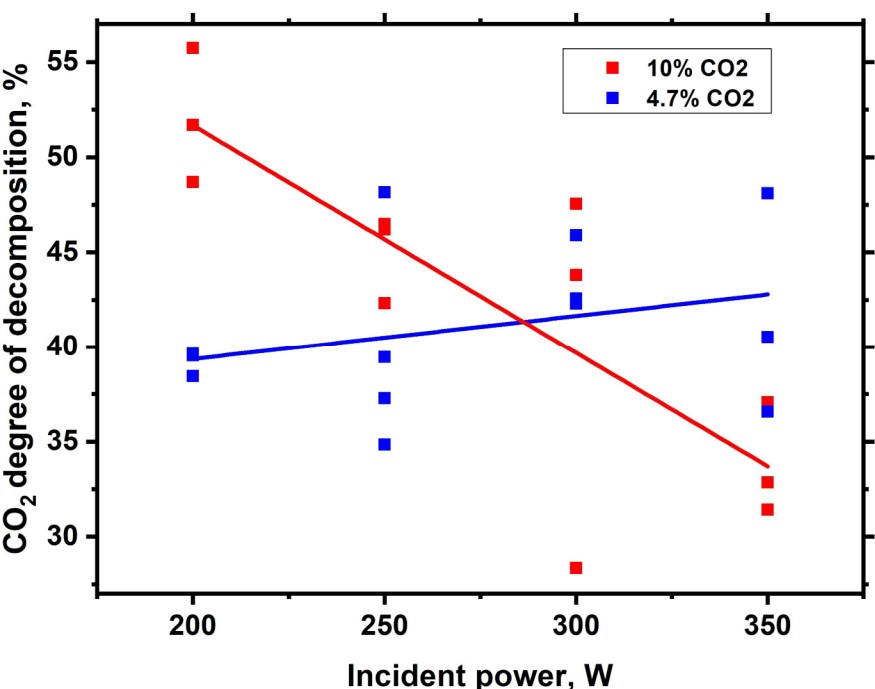

**Figure 7.** Degree of $CO_2$ decomposition against the incident microwave power.

Degree of $CO_2$ decomposition, $\alpha$(%), is calculated by the following formula:

$$\alpha(\%) = \left(1 - \frac{Q^{CO_2}(out)}{Q^{CO_2}(in)}\right) \times 100\% \qquad (3)$$

where $Q^{CO_2}(in)$ and $Q^{CO_2}(out)$ are the flow rates of $CO_2$ [cm$^3$/s] at the inlet and outlet of the plasma reactor.

Figure 7 shows that high degrees of $CO_2$ decomposition have been achieved in the discharge. Comparison with the data given in the review on the decomposition of $CO_2$ in conventional microwave discharges at atmospheric pressure [13], shows that the obtained degrees of decomposition of $CO_2$ correspond to the best indicators known in the literature on microwave discharges. The results shown in Figure 7 open the way for process optimization.

It is rather difficult to determine the energy efficiency of the $CO_2$ decomposition process under the conditions of our experiments. This is due to the fact that the energy deposited in the discharge is spent not only on the decomposition of $CO_2$, but also on the formation of hydrogen and other gas products from Nefras S2 80/120, and it is problematic to determine the distribution of energy between these processes in experiments. If, nevertheless, we assume that all the absorbed energy in the plasma goes to the decomposition of $CO_2$, then the energy cost of the decomposition of $CO_2$ is about 100 eV/mol, and the energy efficiency of the process is 0.5–2% (determines the ratio of the input energy and the bond breaking energy in the $CO_2$ molecule). A comparison of these indicators with the data from [13], shows that they are much worse than the known data, especially in terms of energy cost for $CO_2$ decomposition. This confirms what has been said above, about the illegitimacy of the proposition of attributing all the plasma absorbed energy as the energy loss on the decomposition of $CO_2$ in our experiments.

The efficiency of the plasma-chemical process is determined not only by the efficiency of achieving the desired result (in our case, the decomposition of $CO_2$), but also by the possibility of obtaining associated useful products. This product is hydrogen in our case.

The results of the study showed that the hydrogen content in the gas mixture at the outlet of the reactor is 50–60%, the volumetric rate of hydrogen formation is in the

range of 220–475 mL/min, the energy efficiency of hydrogen formation, calculated from the incident power, is in the range of 65–81 NL/kWh. These values are commensurate with the values obtained by microwave discharge in liquid n-dodecane (1560 mL/min and 74 NL/kWh) [40], and exceed the performance values obtained in DBD in C1–C16 alkanes (17–34 mL/min and 24–121 NL/kWh, where the best results were obtained in n-hexadecane) [41].

## 4. Conclusions

The methods of gas chromatography, optical emission spectroscopy, and shadow photography were used to study the decomposition of $CO_2$ in a microwave discharge, in a liquid petroleum solvent, with a mixture of argon and $CO_2$ bubbling through a channel in the antenna. The pressure in the discharge system was 1 atm, the incident microwave power ranged between 200 and 350 W. The flow rate of the gas mixture was kept constant, at 0.6 L/min, the $CO_2$ flow rate was 10% and 4.7% of the total flow rate (60 and 30 mL/min). The discharge burns in a gas bubble at the end of the microwave antenna; as the discharge burns, the size of the bubble grows, at a rate of about 2.5 m/s. The final diameter of the bubble reaches a size of about 1 cm. An upper estimate of the specific energy input into the plasma gives values in the range of 150–350 $W/cm^3$.

The main gas products are shown to be $H_2$, $C_2H_2$, $C_2H_4$, $CH_4$, $CO_2$, and CO. No oxygen was found in the products; the degree of $CO_2$ decomposition reaches 60%. It was concluded that, at 5% $CO_2$ content at all incident powers, and at 10% $CO_2$ content at lower levels of incident power, the main channel of $CO_2$ decomposition is the reaction $CO_2 + CH_4 \rightarrow 2CO + 2H_2$.

Simultaneously with the decomposition of $CO_2$, hydrogen is generated in the discharge. The results of this study show that the hydrogen content in the gas mixture at the reactor outlet is 50–60%, the volumetric rate of hydrogen formation is in the range of 220–475 mL/min, and the energy efficiency of hydrogen formation, calculated from the incident power, is in the range of 65–81 NL/kWh.

The results obtained show the efficiency of using plasma in liquid hydrocarbons for the decomposition of $CO_2$ with the simultaneous production of hydrogen.

**Author Contributions:** Conceptualization, writing Y.A.L.; formal analysis, I.V.B.; chromatography, T.S.B. and N.A.N.; optical measurements, G.V.K. All authors have read and agreed to the published version of the manuscript.

**Funding:** Russian Science Foundation, grant no. 17-73-30046.

**Informed Consent Statement:** Not applicable.

**Data Availability Statement:** The data presented in this study are available on request from the corresponding author. The data are not publicly available due to restrictions privacy.

**Conflicts of Interest:** The authors declare no conflict of interest.

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
