# Peer review of "CO2 Decomposition in Microwave Discharge Created in Liquid Hydrocarbon"

_plasma, doi:10.3390/plasma6010010_

Round 1

Reviewer 1 Report

Some remarks:

1.     Abbreviation “OES diagnostic” should be introduced.

2.     Figure 2 will be more readable if there is indication for the type of the line for experimental and calculated spectra instead use of numbers. Should be better to insert in the figure caption of additional text, for example, red dot line for calculated spectra and black solid line – experiment. There are many numbers on the figure.  

3.     Proposition for one part of figure caption of Figure3: (1-2-3 indicate evolution of spectrum) instead “(the time increased in the line 1-2-3)”.

4.     Figure 4: Proposition for time presentation of horizontal axis, t (s) – for example.

5.     Page 6, line 186: The text appears twice.

6.     Page 10, line 314: It seems better to move the text (“The pressure ….. (60 and 30 ml/min))  in section “Experimental setup”.

7.     Where are Table 3 and Table 4?

Author Response

We are grateful to the esteemed Reviewer for his interest to our article and his detailed analysis of it. All comments are very helpful in improving the presentation of the information.

All Reviewer’s comments were carefully analyzed and were taken into account in the revised version of the article.

Below the Reviewer’s comments are marked by red italic types and answers are given in black normal letters.

  1. Abbreviation “OES diagnostic” should be introduced.

Thank you. This is our omission. Abbreviation “OES” in the text was replaced by “the optical emission spectroscopy”.

  1. Figure 2 will be more readable if there is indication for the type of the line for experimental and calculated spectra instead use of numbers. Should be better to insert in the figure caption of additional text, for example, red dot line for calculated spectra and black solid line – experiment. There are many numbers on the figure.  

Thank you. We agree with you. Figure 2 was changed and additional text was added to figure caption. New caption is:  Comparison of measured (black solid line) and calculated (red dot line) spectra of Swan bands.

  1. Proposition for one part of figure caption of Figure3: (1-2-3 indicate evolution of spectrum) instead “(the time increased in the line 1-2-3)”.

Thank you. New caption in Fig.3 is “Dynamics of spectrum during on experiment. Numbers near the curves correspond to 179 the spectra measured in different time from the discharge ignition (1-2-3 indicate evolution of spectrum)”

  1. Figure 4: Proposition for time presentation of horizontal axis, t(s) – for example.

Thank you. We agree and horizontal axis in the revised version of Figure 4 is given in time scale

  1. Page 6, line 186: The text appears twice.

Thank you. This is our omission. Text was deleted.

  1. Page 10, line 314: It seems better to move the text (“The pressure ….. (60 and 30 ml/min))  in section “Experimental setup”.

Thank you for suggestion. But this text is already presented in the experiments section. It seems that its repetition here is appropriate for a holistic view of the experimental conditions and results.

  1. Where are Table 3 and Table 4?

Thank you. This is our omission. Text was deleted.

 Hope that answers have satisfied the respected Reviewer.

Reviewer 2 Report

The paper presents new results. It may be published after considering the points listed below.

Line 27: how do you know that it has a “negative” and not a “positive” impact?

Line 43: “The” temperature of “the” liquid does not excedd ….

Line 100 and line 58: Kindly justify the use of Nefas. Why not using any other liquid, e.g., water?

Line 103: which alkanes? How large is “n”?

Figures 3 and 4: the time, e.g, the meaning of 1, 2, and 3 (Figure 3) must be provided. Axis of figure 4 must be a time in sec, min, or hours. This is a “must”. Also in text.

Line 190: what is “space”? volume?  

Line 194” “area” or “region”?

Line 201 “top estimate”?

Figure 5: Indicate diameter (size) of antenna in the figure as a bar. “Bubble” is in the center? What are the object surrounding it? What about using color indicating the “bubble”?

Figures: quality is poor (washy). In my opinion, copied with insufficient resolution.    

Author Response

Reply to Reviewer 2

We are grateful to the esteemed Reviewer for his interest to our article and his detailed analysis of it. All comments are very helpful in improving the presentation of the information.

All Reviewer’s comments were carefully analyzed and were taken into account in the revised version of the article.

Below the Reviewer’s comments are marked by red italic types and answers are given in black normal letters.

Line 27: how do you know that it has a “negative” and not a “positive” impact?

Thank you! Agree. The text is not written clearly. This meant an indirect impact on humans through the process of global warming. Since the text raised a question, this sentence has been changed to: CO2 emissions result from the combustion of fossil fuels (coal, natural gas and oil) in energy and transport.

Line 43: “The” temperature of “the” liquid does not excedd ….

Thank you. Text was corrected as” The temperature of the liquid does….”

Line 100 and line 58: Kindly justify the use of Nefas. Why not using any other liquid, e.g., water?

Thank you.  It is very good question.

Let me remind you that we want to study the decomposition of CO2 in plasma in a liquid in which hydrogen and methane are one of the products. It is known that the presence of hydrogen and methane has a positive effect on the decomposition of carbon dioxide in the plasma.

Such liquids are hydrocarbons. We have extensive experience in the study of microwave discharge in liquid hydrocarbons with carbon content up to C16. It is important that the composition of gas products in all hydrocarbons is approximately the same and contains hydrogen and methane.

Therefore, alkanes can be used in experiments.

Why we didn't use pure individual alkanes but Nefras? The determining factor here was that the result would be the same as in alkanes, but Nefras is much cheaper. Note also that one of the advantages of alkanes is their small loss tangent in the microwave range of wavelengths, so that all the energy is put into the plasma, and not into the heating of the liquid.

Is it possible to use any other liquids, e.g. water?

Water is not good candidate as methane is not among plasma products. (as it is seen from the paper, namely presence of methane gave us possibility to explain the obtained results.).

This difficulty can be overcome by using alcohol solutions. We have a lot of experience with microwave discharge in such liquids, but for other purposes. Their main disadvantage is that, due to the large loss tangent, it is required to operate at power levels of the order of 1 kW, with the bulk of the energy spent on heating and evaporating the liquid.

Line 103: which alkanes? How large is “n”?

Thank you. n<8. This is included in the text.

Figures 3 and 4: the time, e.g, the meaning of 1, 2, and 3 (Figure 3) must be provided. Axis of figure 4 must be a time in sec, min, or hours. This is a “must”. Also in text.

Thank you.

New caption in Fig.3 is “Dynamics of spectrum during on experiment. Numbers near the curves correspond to 179 the spectra measured in different time from the discharge ignition (1-2-3 indicate evolution of spectrum)”

.We agree and horizontal axis in the revised version of Figure 4 is given in time scale

Line 190: what is “space”? volume?  

Thank you. It is our mistake. We delete the text “The results presented in Tables 3 and 4 are important for organizing measurements 189 of the concentration of the gaseous components of the plasma and the space velocity of 190 the gases at the reactor outlet.”

Line 194” “area” or “region”?

Thank you. You are right. The word “area” is replaced by “region” in the revised version.

Line 201 “top estimate”?

Thank you. It is our mistake. The corrected sentence is: “The obtained dimensions allow us to estimate from above the specific energy…”

Figure 5: Indicate diameter (size) of antenna in the figure as a bar. “Bubble” is in the center? What are the object surrounding it? What about using color indicating the “bubble”?

Thanks for the question. Question indicates that the description of Figure 5 in the text is not enough. This description has been added. Figure was changed and now it shows the antenna and bubble diameters. It seems that in the new figure, with the explanation, no color indication of the bubble is required. Moreover, a black and white drawing allows to more accurately determine its border.

A shadow photograph is obtained by irradiating the reactor (a discharge tube with an antenna in the center filled with Nefras) by an external source and shows the change in density inside the reactor.

The added text: “The black rectangle at the bottom of each photo corresponds to the antenna. The bright area at the end of the antenna is the glowing area of the discharge. It is surrounded by a black area, which is an image of a gas bubble, the outer size of which corresponds to its diameter. The luminous region outside the bubble is formed by radiation from an external light source.”

Figures: quality is poor (washy). In my opinion, copied with insufficient resolution.    

Figures were improved.

Hope that answers have satisfied the respected Reviewer.

Reviewer 3 Report

The work is devoted to the current direction of research focused on the utilization of CO2 or its transfer to other compounds. Despite the generally good impression of the work, I have several comments that interfere with the favorable perception of this work.

The title of the work may already mislead the reader, since it may give the impression that with the help of plasma, CO2 is converted INTO liquid hydrocarbons. I suggest that the authors consider the possibility of correcting the title of the work.

Figure 6 shows the "flow rates" for different components. This fundamentally contradicts what is observed - you have ONE stream at the output with a certain speed, in which you can observe some concentrations of compounds. This should be indicated, because no component separation device is presented at the output.

The accuracy of determining the concentrations of CO2 and CO by the presented chromatographic method is low. Therefore, it would be too ambitious to consider the change in the observed values (within the presented limits) as a pattern.

Regardless of the concentration of CO2 in the reaction mixture and the input power, the amount of CO at the output is almost 2 times higher than the incoming CO2. This makes it seem that the plasma interacts more with the solution than with the dissolved gas. In this regard, the question arises about the need for control experiments. In the above results, there is no data on the concentrations of components at the reactor outlet without plasma exposure.

In Experimantals "CO2 flow rate was 10 and 4.7% of the total flow rate (60 and 28.2 ml/min", but in Conclusion "the CO2 flow rate was 10 and 4.7% of the total flow rate (60 and 30 ml/min)." Why do you prefer to present it twice and  at the same time you specify incorrect values at the second mention?

Round 2

Reviewer 2 Report

Paper is improved and now ready for publication. 

Reviewer 3 Report

The responses provided to the comments and corrections made to the text allowed for a better understanding of the work done by the authors and their motivation when presenting the results. Thanks.